# Metal chloride perovskite thin film based interfacial layer for shielding lithium metal from liquid electrolyte

Yi-Chen Yin[1,2,7], Qian Wang[3,7], Jing-Tian Yang[4,7], Feng Li[1], Guozhen Zhang 📧 [1,3✉], Chen-Hui Jiang[5], Hong-Sheng Mo[2], Ji-Song Yao[2], Kun-Hua Wang[2], Fei Zhou 📧 [1], Huan-Xin Ju 📧 [6✉] & Hong-Bin Yao[1,2✉]

Fabricating a robust interfacial layer on the lithium metal anode to isolate it from liquid electrolyte is vital to restrain the rapid degradation of a lithium metal battery. Here, we report that the solution-processed metal chloride perovskite thin film can be coated onto the lithium metal surface as a robust interfacial layer to shield the lithium metal from liquid electrolyte. Via phase analysis and density functional theory calculations, we demonstrate that the perovskite layer can allow fast lithium ion shuttle under a low energy barrier of 0.45 eV without the collapse of framework. Such perovskite modification can realize stable cycling of LiCoO$_2$|Li cells with an areal capacity of 2.8 mAh cm$^{-2}$ using thin lithium metal foil (50 μm) and limited electrolyte (20 μl mAh$^{-1}$) for over 100 cycles at 0.5 C. The metal chloride perovskite protection strategy could open a promising avenue for advanced lithium metal batteries.

[1] Hefei National Laboratory for Physical Sciences at the Microscale, University of Science and Technology of China, 230026 Hefei, Anhui, China. [2] Department of Applied Chemistry, University of Science and Technology of China, 230026 Hefei, Anhui, China. [3] Department of Chemical Physics, University of Science and Technology of China, 230026 Hefei, Anhui, China. [4] School of the Gifted Young, University of Science and Technology of China, 230026 Hefei, Anhui, China. [5] CAS Key Laboratory of Materials for Energy Conversion, Department of Materials Science and Engineering, University of Science and Technology of China, 230026 Hefei, Anhui, China. [6] PHI China Analytical Laboratory, CoreTech Integrated Limited, 402 Yinfu Road, 211102 Nanjing, China. [8] These authors contributed equally: Yi-Chen Yin, Qian Wang, Jing-Tian Yang. ✉email: guozhen@ustc.edu.cn; huanxin.ju@coretechint.com; yhb@ustc.edu.cn

Rapidly developing electronic devices provide us with better product experience, but simultaneously set a higher standard for energy storage systems[1]. It stimulates extensive search of new materials to meet the increasing demands of high-energy densities of rechargeable batteries[2–4]. With 10 times the specific capacity of graphite (3860 versus 370 mAh g⁻¹), metallic lithium (Li) is among the most promising candidates for future anode to support high-energy density batteries[5,6]. However, the practical application of Li metal batteries is hindered by a major issue associated with loose structure of deposited Li during cycling, which would result in accelerated electrolyte "dry-out" failure and probably short circuit of the battery[5,7].

The solid electrolyte interphase (SEI) layer on the surface of Li metal anode plays a predominate role in controlling the deposition of Li metal[8,9]. In the early stage to reinforce the SEI layer, various electrolyte additives, including vinylene carbonate[10], LiNO₃[11], and fluorinated ethylene carbonate[12], have been tried, but the obtained mechanical strength is not enough to suppress the growth of Li dendrites. Lately, the fabrication of artificial SEI protection layer via pre-treatment methodology has been proposed. A series of artificial SEI protection layers, including carbon spheres[13], polymers[14,15], alloy layer[16], composite films[17,18] have been applied on the surface of Li metal anodes. Despite these artificial SEI layers exhibited the capability to suppress the growth of Li dendrites[16,19], the ultra-dense deposition of Li along with the suppressed detrimental side reactions with liquid electrolyte have not been achieved due to the permeation of liquid electrolyte throughout the protection layers.

Directly fabricating a highly ionic conductive dense thin film on the surface of Li metal anode is a promising route to fully shield the Li metal from liquid electrolyte[7,20]. To design a dense interfacial layer for Li⁺ ion conduction, the framework structure of material should own continuous channels for Li⁺ ion migration[21,22]. The perovskite structure with the general formula ABO₃ (where A is a cation and B is a divalent metal ion) provides a highly symmetric host to generate a classical Li⁺ ion conductor, lithium lanthanum titanate (LLTO)[23], but the fabrication of this perovskite host needs high temperature sintering[24,25], which hindered the application of LLTO thin film on the surface of Li metal anode. Similar to the LLTO perovskite solid electrolyte, other ceramic Li⁺ ion conductors with high shear modulus are also very challenging to be coated as thin films on the Li metal anodes due to their critical preparation conditions[26,27].

As a new kind of perovskite, metal halide perovskite with a general formula ABX₃ (X is a halide) can be produced as the form of thin film via a facile solution-processable method[28–30]. The metal halide perovskites have shown high photoelectric conversion efficiency in planar solar cells and attracted intensive attention in the past decade[31]. Aside from their unique photo-electronic properties, the framework feature of metal halide perovskite can realize the Li⁺ ion intercalation[32,33], indicating its potential as a fast Li⁺ ion transport material. In addition, to be an electronic insulating layer, metal halide perovskites can get its bandgaps enlarged by using chloride ions as the halide sources[34]. These features of metal chloride perovskite make it as an attractive interfacial material with high Li⁺ ion transport capability to isolate the Li metal anode from liquid electrolyte.

Herein, we develop a solid-state transfer process to apply solution processed metal chloride perovskite thin film as a new type of interfacial layer onto the Li metal anode. The metal chloride perovskite interfacial layer can allow fast Li⁺ ion transport throughout its framework with a low energy barrier, endowing homogenous ion flux with excellent isolation from liquid electrolyte. Thus, the ultra-dense deposition of Li with high areal capacity (30 mAh cm⁻²) induced by a high current density of 5 mA cm⁻² is demonstrated and the Li metal anode under the protection of metal chloride perovskite exhibits a stable Li stripping/plating voltage curve (<40 mV) at the current density of 1 mA cm⁻² for over 800 h. Attractively, with lean electrolyte (20 μl mAh⁻¹) and thin Li foil (50 μm), the LiCoO₂-Li cell with an areal capacity of 2.8 mAh cm⁻² can be cycled for more than 100 cycles.

## Results

**Shielding mechanism of metal chloride perovskite.** Based on the intrinsic feature of metal chloride perovskite, we propose a fast Li⁺ ion transport gradient layer model to illustrate the shielding mechanism of perovskite thin film for the dense deposition of Li metal (Fig. 1a). At first, the perovskite framework can only allow the entrance of Li⁺ ions, leaving the solvent molecules out, when a negative electric field is applied, because the size of perovskite cavity can only accommodate desolvated Li⁺ ions. After intercalation into the framework, the Li⁺ ions can easily migrate along the contentious vertical channels due to the high symmetry of the perovskite framework. Once the Li⁺ ions approach the interface of perovskite layer and the substrate, a conversion-type electrochemical reaction will occur to form the Li–M alloy layer and the insulated LiCl layer. The Li–M layer will promote a homogeneous consequent deposition of Li, which has been confirmed in previous reports[16,19]. In addition, the generated LiCl can insulate electrons and prevent the perovskite from further conversion reaction, retaining the stability of the perovskite on the top surface. Therefore, the finally formed solid electrolyte interfacial layer is a gradient thin film consisting of perovskite framework on the top surface to isolate the liquid electrolyte and underneath Li–M alloy layer to induce the homogeneous deposition of Li metal.

To unravel the conduction mechanism of Li⁺ ion in the perovskite lattice, we performed density functional theory (DFT) calculation to simulate the migration of Li⁺ ions in the perovskite framework (Supplementary Note 1), obtaining the specific migration pathways and associated energy barriers (Supplementary Fig. 1). The simulated model is based on the cubic phase MASnCl₃ (MA = methylammonium, CH₃NH₃ in formula) and the potential energy surface along the migration pathway of Li⁺ ion is shown in Fig. 1b. The corresponding positions of Li⁺ ion at its initial state, transition state and final state are displayed as the insets. To accommodate the migration of Li⁺ ions, the lattice undergoes mild yet measurable distortion, accompanied by the orientation adjustments of MA ion due to the nonuniform distribution of electric potential in the lattice (Supplementary Fig. 2). During the diffusion process, the transition state at the peak of potential energy curve is reached as the Li⁺ ion is about to pass through the shared face of two neighboring unit cells of perovskite. Interestingly, the MA ion continues adjusting its orientation along with the migration of Li⁺ ion and restores its original direction as Li⁺ ion enters another unit cell. This is crucial for Li⁺ ion to maintaining a repeatable migration in the perovskite lattice and enabling the Li⁺ ion conduction throughout the perovskite framework. The estimated energy barrier is 0.45 eV, close to the activation energy barrier of a couple of other known solid-state-electrolytes (e.g., 0.53 eV for Li₄GeS₄, 0.49 eV for γ-Li₃PS₄, to name a few)[35], indicating the ease of Li⁺ ion conduction in the metal chloride perovskite framework.

To demonstrate the gradient layer model of the perovskite, we fabricated the metal chloride perovskite thin films composed of MASnCl₃ or MAPbCl₃ with high film quality on the substrates via solution spin-coating technique at first (see Methods and Supplementary Fig. 3). The thickness of as-fabricated perovskite film is ~1 μm for MASnCl₃ and ~2 μm for MAPbCl₃ (Supplementary Fig. 4). The X-ray diffraction (XRD) patterns (Supplementary Fig. 5) show that the pure perovskite thin films possess

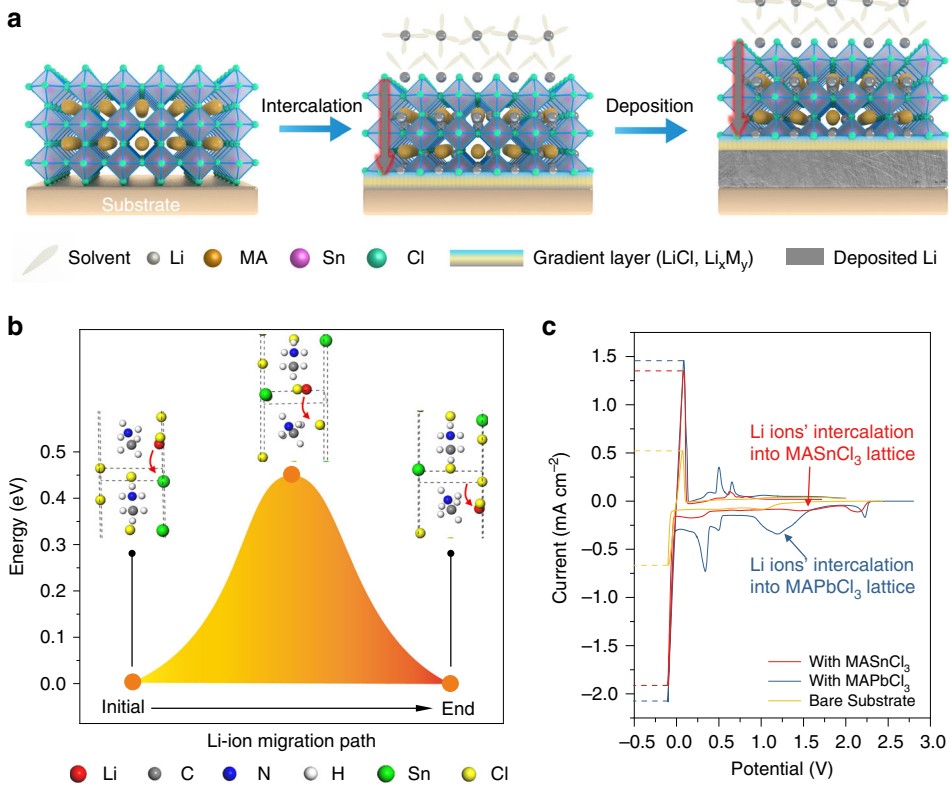

**Fig. 1 Mechanism exploration of Li$^+$ ion migration through the lattice of metal chloride perovskite for shielding. a** Schematic illustration of the mechanism of Li ions' intercalation into perovskite lattice, the formation of perovskite-alloy gradient Li ion conductor and the deposition process. **b** hypothetical migration pathway of Li$^+$ ion and corresponding potential energy surface highlighting the initial state, peak state and final state in a period. **c**, Cyclic voltammetry comparison of the cells using MASnCl$_3$-coated, MAPbCl$_3$-coated or bare substrate as the working electrode, and the lithium as the counter and reference electrode with the sweep rate of 1 mV s$^{-1}$.

highly oriented crystalline structure with a series of diffraction peaks belonging to crystal indices of (n 0 0) (n = 1, 2, 3…). Such highly oriented crystalline structure is consistent with our proposed framework structure in Fig. 1a, which will facilitate the Li$^+$ ion transport throughout the perovskite layer.

We investigated the cyclic voltammetry (CV) of Li plating on these thin film-coated substrates and compared them to that of bare substrate. As shown in Fig. 1c, the perovskite film coated substrates experience an intercalation of Li$^+$ ions appearing at the potential peak higher than 1 V (~1.5 V for MASnCl$_3$, and ~1.2 V for MAPbCl$_3$) in comparison to the flat current curve in the CV of bare substrate without the characteristic Li$^+$ ion intercalation process. The other CV peaks with the potential around 0.4–0.7 V belong to the reduction process of interfacial perovskite to Li-Sn or Li-Pb alloy, and the final state of as-formed alloy is Li$_{17}$Sn$_4$ or Li$_{17}$Pb$_4$[36–38]. As the potential approaches to −0.1 V, the current density of the substrate with MASnCl$_3$ or MAPbCl$_3$ is around 1.5 mA cm$^{-2}$, much higher than the current density of bare substrate (~0.6 mA cm$^{-2}$). Similarly, at the symmetric voltage position near +0.1 V, the perovskite-modified substrates also present much higher response currents, indicating the promotion of Li plating/stripping kinetics by the perovskite thin films. To further show the unique role played by the perovskite framework rather than as-formed Li–M alloy layer, we compared the CV curves of MAPbCl$_3$ and PbCl$_2$-coated substrates (Supplementary Fig. 6). The absence of typical Li$^+$ ion intercalation peak around 1.2 V and much lower Li plating/stripping current density of PbCl$_2$-coated substrate in comparison to that of MAPbCl$_3$ coated

substrate demonstrated the unique function of perovskite structure to promote Li plating/striping kinetics.

To confirm the self-prevented electrochemical reduction of perovskites, we evaluated the amount of electrochemically reduced perovskites through galvanostatic lithiation tests (Supplementary Fig. 7). The calculated results (Supplementary Note 2) showed that only a small part of perovskites (~30 % for MASnCl$_3$ and 15 % for MAPbCl$_3$), corresponding to the absolute quantities (about 2.8 × 10$^{-7}$ mol cm$^{-2}$, or 0.3 μm in thickness) of perovskites, was reduced to form Li$_{17}$Sn$_4$ or Li$_{17}$Pb$_4$ layer. The limited perovskite consumption independent of the original thickness values of perovskite layer implies that the perovskite thin films are not completely reduced due to the formation of insulating LiCl layer, as well as the intrinsic property of wide band gap of metal chloride perovskite to isolate electrons coming from the current collector. Therefore, the finally formed gradient interfacial layer is consisted of perovskite framework on the top surface followed by the Li–M alloy/LiCl layer beneath, which is evidenced by the depth X-ray photoelectron spectroscopy (XPS) analysis (Supplementary Fig. 8 and Supplementary Note 3).

**Ultra-dense Li deposition underneath metal chloride perovskite.** To demonstrate practical protection role of metal chloride perovskite, we investigated the Li plating behavior underneath these metal chloride perovskite thin films. We directly applied Li plating onto the perovskite-coated substrates in half cells (see Methods and Supplementary Fig. 9a). As shown in Fig. 2a, after

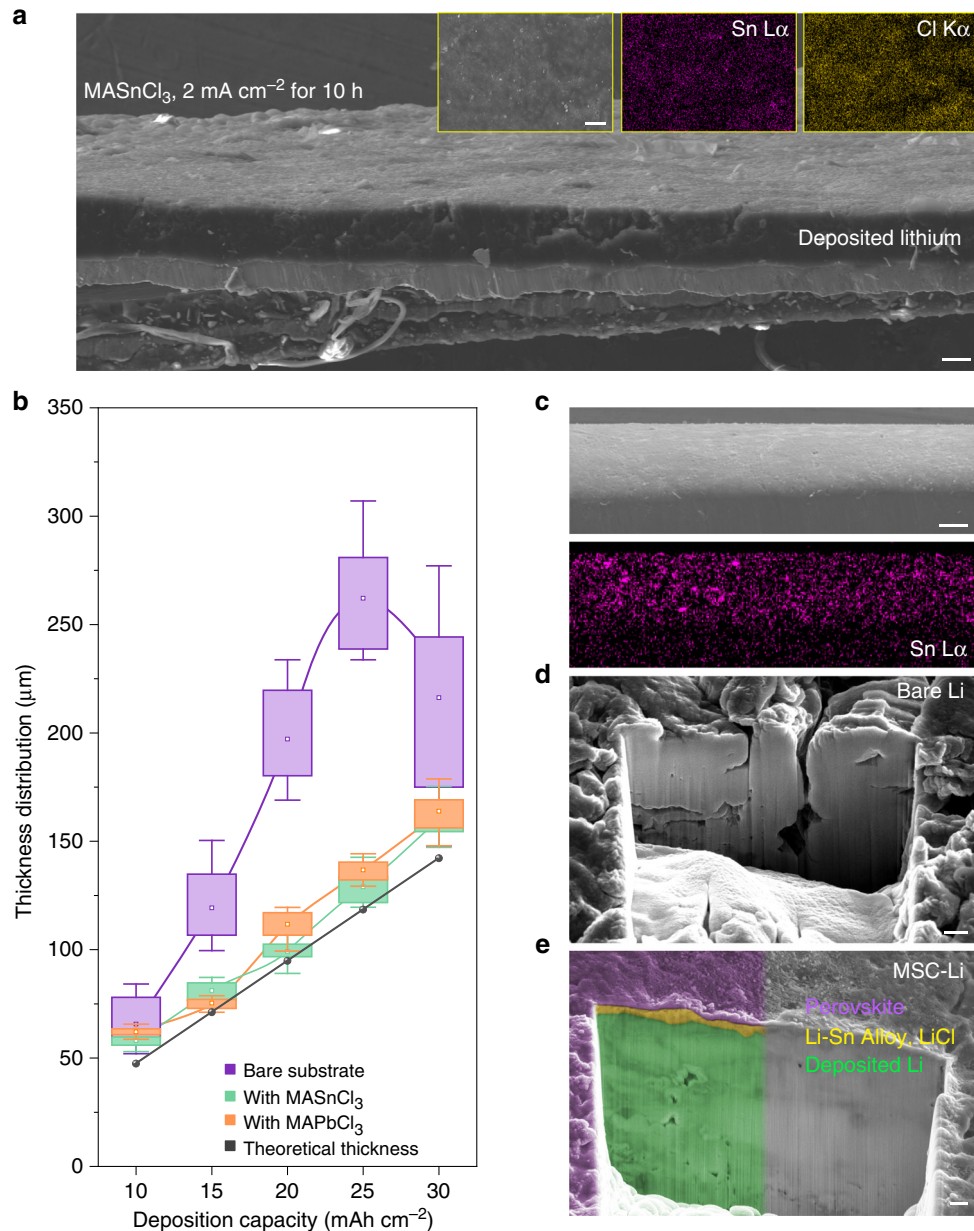

**Fig. 2 Ultra-dense Li deposition enabled by high-quality perovskite thin film. a** Morphologies of Li electrodeposited on the MASnCl$_3$-coated substrate, and the SEM image of the top surface with corresponding EDX mappings indicating the distribution of Sn, Cl on the top surface of as-deposited Li. **b** Thickness statistics of Li electrodeposited on the bare (purple), MASnCl$_3$-coated (green) or MAPbCl$_3$-coated (orange) substrate. Theoretical thickness of Li is set as a reference (black). Current densities range from 2 to 6 mA cm$^{-2}$ and the deposition time lasted for 5 h. **c** Side-view SEM image and corresponding EDX mapping indicating the distribution of Sn on the surface of MSC-Li. **d e** FIB-processed deep cross-sectional morphologies of bare Li (**d**) and MSC-Li (**e**) after electrodeposition under 4 mA cm$^{-2}$ for 2 h, where the different color based on the contrast difference implies the retention of perovskite-alloy-lithium (purple–yellow–green) gradient structure of MSC-Li. Scale bars: 50 μm in a, 100 μm in c, and 1 μm in **d**, **e**.

the Li plating under 2 mA cm$^{-2}$ for 10 h, an ultra-dense and dendrite-free Li metal deposition is observed underneath the MASnCl$_3$ perovskite thin film. The energy dispersive X-ray spectroscopy (EDX) analyses (Insets in Fig. 2a) confirmed that the top surface of the substrate after the Li plating is still composed of Sn and Cl, which indicates that the Li plating was processed underneath the perovskite thin film. In order to confirm that the perovskite film was not completely reduced during electrochemical Li deposition, XPS was performed to examine the chemical state of Sn (Supplementary Fig. 9b). After Li deposition, the Sn 3d spectrum of MASnCl$_3$-coated substrate shows higher binding energy peaks of oxidation state of Sn than that of zero valence state Sn$^0$, indicating no reduction of MASnCl$_3$ on the top

surface. Moreover, XRD pattern of the deposited Li on MASnCl$_3$-coated substrate confirmed the obtained lithiated perovskite with highly oriented crystal structure (Supplementary Fig. 10b), implying that the formed lithiated perovskite thin film inherited the highly oriented crystal structure from original MASnCl$_3$ thin film. The similar dense Li deposition behavior can be also achieved underneath the MAPbCl$_3$ perovskite thin film with retained high crystal orientation (Supplementary Figs. 9c, 10c, and 11).

We further carried out the Li plating on MASnCl$_3$ or MAPbCl$_3$ coated substrates under the current densities from 2 mA cm$^{-2}$ to 6 mA cm$^{-2}$ for 5 h. The dense structure of plated Li metal can be observed underneath all metal chloride perovskite layers even under the current density up to 6 mA cm$^{-2}$ (Supplementary

Fig. 12). The top surfaces of all perovskite-modified Li deposits exhibit smooth appearance without any dendrites (Supplementary Fig. 13). In contrast, the Li plated on the bare substrate presents a highly porous structure and obvious dendrites. We measured the thickness of the plated Li and correlated it with the areal capacity of Li as plated (Supplementary Tables 1 and 2). As shown in Fig. 2b, the thickness of Li deposited underneath metal chloride perovskite films shows nearly the same linear relationship with the areal capacity as that of theoretical value of 100% dense Li (Supplementary Note 4). In contrast, the Li plated on the bare substrate exhibits exponential increase of thickness with the area capacity. The abnormal decline of the thickness of Li deposited on the bare substrate at 6 mA cm$^{-2}$ can be attributed to the serious dendrite growth and pulverization causing the loss of "dead Li" in the electrolyte.

To apply the perovskite thin film onto practical Li metal anode, we developed a solid-state pressing process to transfer perovskite thin film from the substrate onto the surface of Li metal (see Methods and Supplementary Fig. 14). As-transferred perovskite thin film can build up an intimate contact with Li metal via the in situ formation of gradient perovskite/Li–M alloy layer (Supplementary Note 5, M = Sn or Pb), accompanied by spontaneous lithiation of perovskite (Supplementary Fig. 15 and Supplementary Note 6). The unique gradient structure of the as-fabricated perovskite/Li–M alloy layer was evidenced by depth analysis of XPS (Supplementary Figs. 16, 17, and 18 and Supplementary Note 7). The solid-state transfer procedure is well compatible with traditional electrode processing, which has the potential to be applied in large scale (Supplementary Fig. 19). The fabricated MASnCl$_3$-coated Li metal anode and MAPbCl$_3$-coated Li metal anode are here denoted as MSC-Li and MPC-Li, respectively. As shown in Fig. 2c and Supplementary Fig. 20, the MASnCl$_3$ film and MAPbCl$_3$ film are tightly and uniformly coated on the surface of Li metal anodes.

Focused ion beam (FIB) was performed to deeply expose the cross-sectional morphologies of bare Li and MSC-Li after 2 h Li plating under a current density of 4 mA cm$^{-2}$. Dense structure of Li deposit is observed in the MSC-Li anode in comparison to the porous structure of that of bare Li metal anode (Fig. 2d, e), implying that the transferred MASnCl$_3$ layer can well induce ultra-dense Li deposition. Notably, MSC-Li exhibits an obvious contrast difference (denoted by purple, yellow and green) in vertical direction near the surface, implying a perovskite-alloy-lithium interfacial layer on the Li metal anode surface to resist the liquid electrolyte corrosion. Similarly, the ultra-dense Li deposition can be also realized on the MPC-Li anode with the retention of perovskite protection layer (Supplementary Fig. 21). In addition, optically transparent cell tests (experiment details are shown in Supplementary Fig. 22 and Methods) also indicated that the Li dendrite growth can be well suppressed by the perovskite-modified layer in comparison to unmodified Li metal anode (Supplementary Fig. 23).

**Stability of metal chloride perovskite coated Li metal anode**. To evaluate the stability of perovskite coated Li metal anode, we carried out the galvanostatic Li plating/stripping test on the symmetric cell. The impedance spectra of symmetric cells using different electrodes before cycling are firstly compared. As shown in Fig. 3a, much lower interfacial resistance values of the cells with MSC-Li (~60 Ω) or MPC-Li (~85 Ω) are obtained in comparison to that of the cell with bare Li electrode (~320 Ω), indicating the stability of perovskite interlayer to promote the charge transfer at the electrode interface. In the following cycling test (Fig. 3b), the MSC-Li cell can be cycled for more than 800 h, much longer than the 100 h cycling life of the cell using bare Li.

The detailed voltage profiles show that the stable overpotential with a low value of 40 mV can be maintained in the cell using MSC-Li electrode. The bare Li electrode, however, exhibits a drastic voltage increase at 100 h, which can be attributed to poor Li$^+$ ion transfer kinetics caused by as formed nonuniform SEI layer. The symmetric cell of MPC-Li electrodes also shows attractive advantages with low overpotential of around 40 mV and a stable cycling life of over 600 h (Supplementary Fig. 24).

Furthermore, we applied XRD analysis to validate the framework stability of highly oriented perovskite layer on the Li metal anode during the cycling. As shown in Fig. 3c, before cycling, the XRD pattern of MSC-Li (blue curve) displayed a series of peaks with indices of (100) and (200) planes of the lithiated perovskite. The other peaks in the XRD pattern can be assigned to Li (marked with "o"). More attractively, after 100 cycles, the MSC-Li still retains a fine correspondence of the XRD pattern (red curve) as that before cycling, demonstrating that the perovskite framework can maintain its structural orientation and behave good stability against the electrolyte corrosion under the electrochemical operation. Similarly, the perovskite framework on the MPC-Li electrode can also keep the phase stability during the electrochemical cycling (Supplementary Fig. 25), further showing the robustness of metal chloride perovskite as the solid interphase layer. By comparing to raw spin-coated perovskite, the as-fabricated lithiated perovskites exhibit no obvious XRD peak shift (Supplementary Fig. 26), implying the good capability of perovskite framework to accommodate Li$^+$ ions. The phenomenon is in good agreement with previous works in which the doping and intercalation of Li$^+$ ions into perovskite framework did not cause significant change in crystal unit cell size[39,40] (Supplementary Fig. 27).

To further explore chemical stability of perovskite layer on the Li metal anode, we conducted XPS analysis on the top surface of MSC-Li electrode before and after cycling. The Sn 3$d$ XPS spectra of MSC-Li are shown in Fig. 3d, where the reference metallic Sn 3$d_{5/2}$ is marked by dotted line at the binding energy of 484.8 eV. Before cycling, the Sn 3$d_{5/2}$ peak is found to be at 486.6 eV, which belongs to the oxidation state of Sn, obviously different from that of metallic Sn. After 100 cycles, it is obvious that the XPS peak of Sn 3$d$ stayed at the same position (486.6 eV for Sn 3$d_{5/2}$), which indicates the chemical stability of perovskite on the top surface of MSC-Li and high consistence with the XRD result. The XPS result is similar for MPC-Li electrode, showing that the lead on the top surface of MPC-Li electrode always retained the oxidation state before and after cycling (Supplementary Fig. 28). All these results indicate that the perovskite layer on the top surface of the Li metal was not reduced to the alloy layer during the cell cycling.

In addition, the stabilities of the perovskite modified electrodes are revealed by the surface morphologies analysis after 100 h cycling. The MSC-Li electrode displays highly dense surface morphology with silvery shining appearance after cycling, in contrast to the drastic roughness and black surface of bare Li electrode (Supplementary Fig. 29). The EDX mapping images of oxygen distribution on the surface of the MPC-Li electrode before and after cycling further reveal that the perovskite film is still dense to endow Li metal with the resistance against oxidation under the exposure to air during the sample transfer for SEM characterization (Supplementary Fig. 30). The EIS curves of the above symmetric cells after cycling in Supplementary Fig. 31 also confirm the advantage of the perovskite modification to facilitate Li$^+$ ion transport.

**Performance of Li metal battery protected by metal chloride perovskite**. To demonstrate the efficiency of perovskite protection for Li metal batteries, we tested the electrochemical performance of Li$_4$Ti$_5$O$_{12}$ (LTO)/perovskite coated Li cells at a high rate

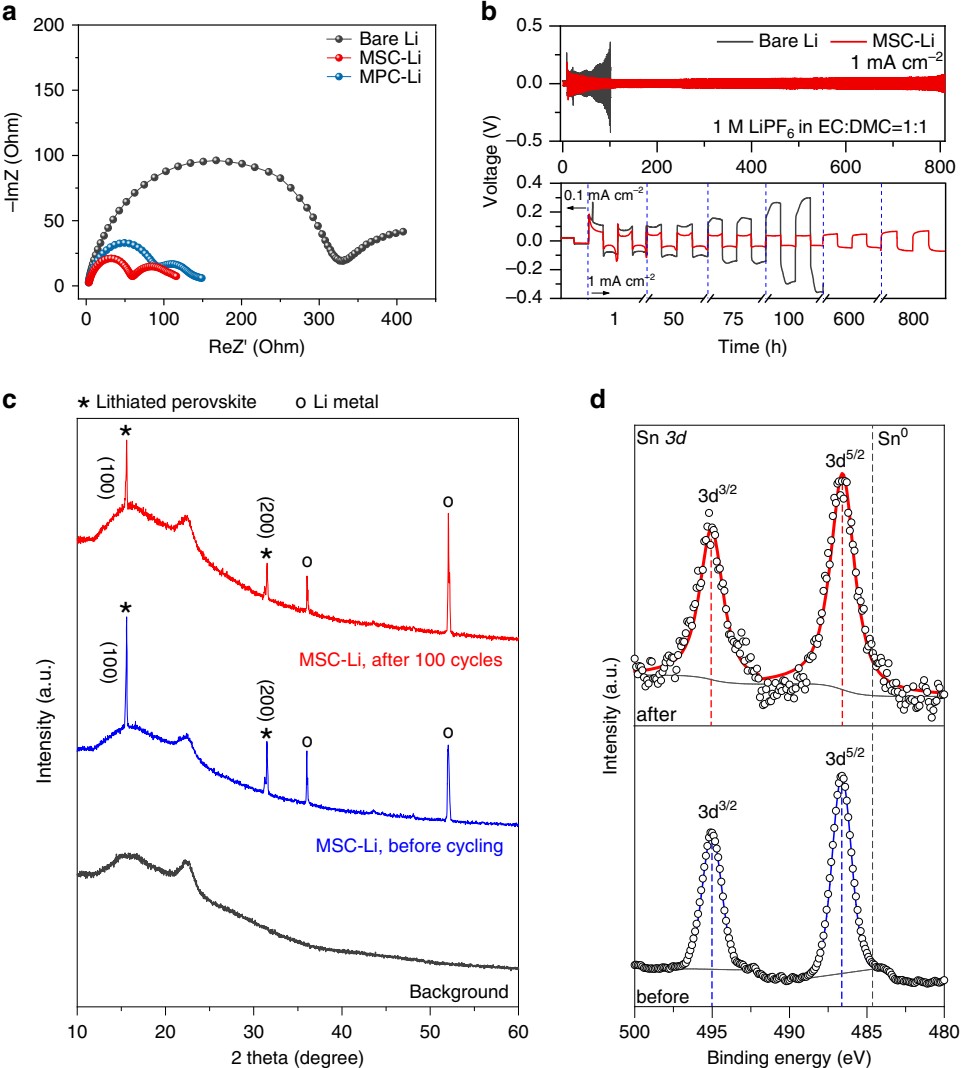

**Fig. 3 Electrochemical stability of MSC-Li and MPC-Li. a** EIS spectra of bare Li, MSC-Li and MPC-Li. **b** Galvanostatic voltage curves (top) of bare Li and MSC-Li tested with a current density of 1 mA cm$^{-2}$ for 1 mAh cm$^{-2}$, and the enlarged voltage curves during different periods (bottom). **c** XRD patterns of MSC-Li anode before cycling (blue) and after 100 cycles (red) to indicate the electrochemical stability and phase retention of lithiated perovskite during cycling, accompanied with the XRD pattern of sealing tape to isolate samples from air (black). **d** XPS spectra of the surface of MSC-Li anode before cycling (blue) and after 100 cycles (red) to prove that the perovskite solid-electrolyte interface stays on the top surface after cycling.

of 5 C. As shown in Fig. 4a, the cell with bare Li metal anode exhibits a drastic capacity decay and the capacity decreased to 28.3 mAh g$^{-1}$ at 400th cycle. In contrast, the cells using MSC-Li and MPC-Li anode show stable cycling for more than 500 cycles with a low capacity decay rate of 0.07 % per cycle. The charge/discharge voltage profiles in Fig. 4b, c show that the charge voltage plateau of the cell using bare Li metal anode is much higher than that of MSC-Li anode, indicating that the perovskite layer promotes the kinetics of Li plating. The discharge voltage plateau of the cell using bare Li anode becomes lower and lower during cycling in comparison to the stable discharge voltage plateau of the cell with MSC-Li, implying a higher resistance for Li stripping caused by deteriorated surface of bare Li anode. The surface SEM images of Li anodes were taken at 400th cycle to further show the protection role played by the perovskite thin films (Fig. 4 d–f). Severe pulverization and loose morphology were observed on the surface of bare Li anode, whereas the surface of MSC-Li or MPC-Li anode still exhibited typical smooth morphologies, demonstrating the robustness of perovskite protection layer in practical Li metal batteries. The impedance spectra of the three cells before

and after 400 cycles are compared in Supplementary Fig. 32, also indicating a stabilized interface of MSC-Li and MPC-Li compared to bare Li metal anode. Additionally, the surface SEM image and corresponding EDS mapping of MPC-Li in Li‖LTO cell after 500 cycles were also taken. As shown in Supplementary Fig. 33, most surface area of MPC-Li is smooth and dense after cycling, revealing good stability of the perovskite layer.

We further explored the performance of perovskite protected Li metal battery by applying strict conditions including high-areal capacity cathode (2.8 mAh cm$^{-2}$), limited Li source (50 μm, equal to 10 mAh cm$^{-2}$) and lean electrolyte (20 μl mAh$^{-1}$) in the LiCoO$_2$|Li cell at 0.5 C. As shown in Fig. 4g, the cell using bare Li metal anode exhibits a rapid capacity decay more than 40% after only 20 cycles. By contrast, MSC-Li and MPC-Li anodes enable the cells to maintain a stable capacity performance for over 100 cycles with a capacity retention of 85%. It indicates that perovskite protective layers can effectively isolate the Li metal from the liquid electrolyte to restrain Li loss and side reactions. In addition, a much lower polarization effect was also observed in the discharge/charge voltage profile of the cell using perovskite

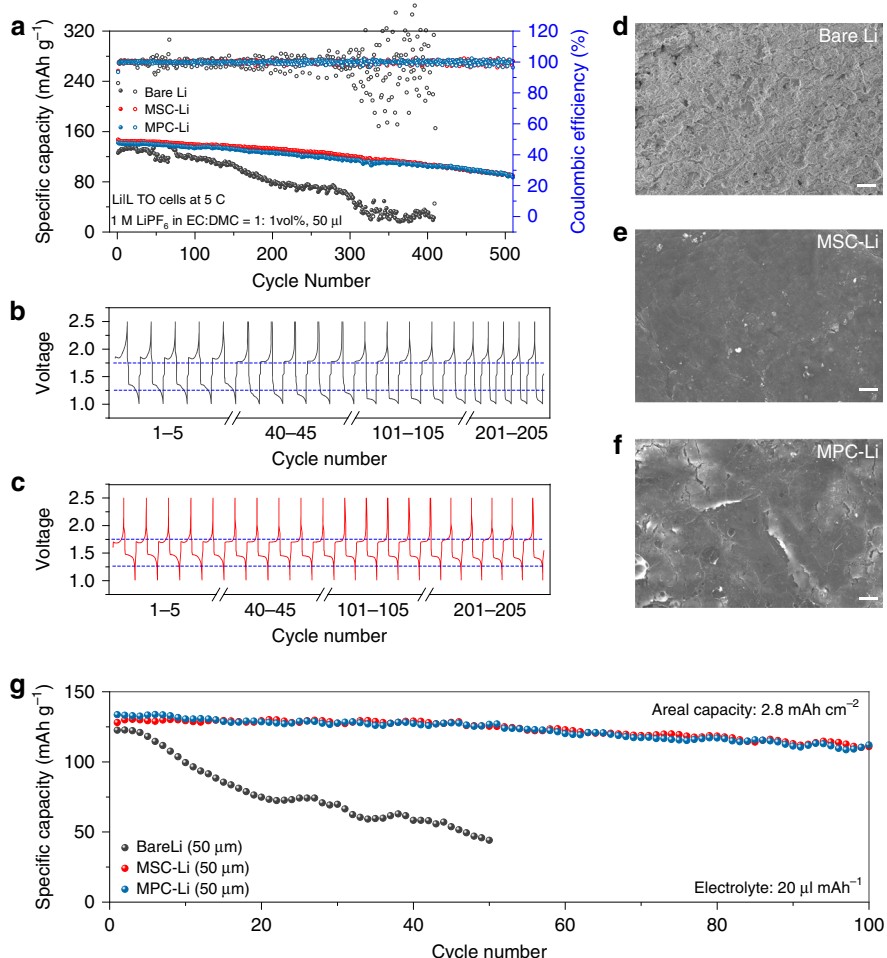

**Fig. 4 Electrochemical performances of Li|LTO cells at 5 C and Li|LCO cells at 0.5 C. a** Galvanostatic cycling performances of cells using Li$_4$Ti$_5$O$_{12}$ as cathode and bare Li, MSC-Li or MPC-Li as anode. **b, c** Voltage curves of bare Li (**b**) and MSC-Li (**c**) at different cycles. **d–f**, Surface morphologies of bare Li (**d**), MSC-Li (**e**), and MPC-Li (**f**) anodes after 400 cycles. **g** Galvanostatic cycling performances of cells using LiCoO$_2$ as cathode and 50 μm thick bare Li, MSC-Li or MPC-Li as anode. Scale bars: 50 μm.

modified Li anode (Supplementary Fig. 34), further demonstrating the good protection of the perovskite thin film for the Li metal anode.

## Discussion

In summary, we have demonstrated that metal chloride perovskite thin films can be used as stable interfacial layers to stabilize the Li metal anode. Due to the electrochemical stability and highly oriented framework for fast Li$^+$ ion conduction, the metal chloride perovskite thin film can not only act as efficient solid electrolyte interlayer to induce ultra-dense deposition of Li underneath their protection, but also isolate the Li metal anode from liquid electrolyte to provide a favorable environment for stable Li plating/stripping, effectively restraining the loss of lithium and the electrolyte during cycling. Furthermore, the metal chloride perovskite protection layer can enable stable cycling of Li metal battery under strict conditions. This innovative strategy to apply solution-processed metal chloride perovskite thin films as interfacial layers on the Li metal anodes will inspire more exploration of metal halide perovskites as ion transport materials, pushing ahead the development of advanced energy storage systems.

## Methods

**Fabrication of metal chloride perovskite thin films**. Stainless steel foil (50 or 100 μm) was washed with deionized water, acetone and ethanol in an ultrasonic cleaner (SCIENTZ, SB25-12DTD, 40k HZ) sequentially before drying in an oven at 70 °C for 30 min. For MASnCl$_3$ perovskite thin films, 0.338 g of methylamine hydrochloride (MACl, 98.5 %, Macklin) and 0.948 g of stannous(II) chloride (SnCl$_2$, 99.9%, Aladdin) were dissolved in 10 mL of N, N-dimethylformamide (DMF, 99.9%, Aladdin) to form MASnCl$_3$ precursor solution with a concentration of 0.5 M. 0.676 g of methylamine hydrochloride (MACl, 98.5%, Macklin). For MAPbCl$_3$ perovskite thin films, 2.781 g of Lead chloride (PbCl$_2$, 99.9%, Aladdin) were dissolved in 10 mL dimethyl sulfoxide (DMSO, 99.9%, Aladdin) to form MAPbCl$_3$ solution with a concentration of 1 M. Each type of as-prepared perovskite precursor solution was spun-coated onto UV–ozone treated stainless steel foil substrate (50 μl for every 1 cm$^{-2}$ of substrate area) at 2000 r/min for 90 s. Then the perovskite coated stainless steel foils were annealed at 70 °C for 5 min. All the MASnCl$_3$ and MAPbCl$_3$ thin films were fabricated and stored in an argon-filled glove box with <1 ppm O$_2$ and <1 ppm H$_2$O.

**Solid-state transfer process to form MSC-Li and MPC-Li anodes**. Li metal foil (99.9%, China Energy Lithium Co., Ltd.) was polished to remove the oxidation layer to obtain a smooth and shinning surface. Then, the as-treated Li foil was pressed onto the perovskite-coated substrate with a pressure of 10 N cm$^{-2}$ for 10 s. The perovskite thin film can be easily transferred from the stainless steel substrate onto the surface of Li metal after separating the Li foil from the stainless steel substrate. MSC-Li and MPC-Li electrodes are thus obtained. The transfer operations were performed at the room temperature of 25 °C in the glove box with <1 ppm O$_2$ and <1 ppm H$_2$O.

**Material characterizations**. XRD measurements were carried out on a X-PERTPRO powder X-ray diffractometer operating at 40 kV and 30 mA with a scanning step of 0.02 degree, using Cu K radiation ($\lambda = 0.15406$ nm). The samples were tested in the Kapton tape-sealed devices for the XRD characterizations (Supplementary Fig. 10a). The micro-morphologies were observed with SU8220 (HITACHI UHR FE-SEM SU8200 Series) with an accelerating voltage of 3 kV for SEM image capture and 20 kV for EDX mapping. All the Li metal contained electrodes were sealed in a transfer vessel filled with argon before being transferred into the chamber of PHI 5000 VersaProbeIII for XPS analysis. Optically transparent cells for in situ observation were assembled using MPC-Li or bare Li as the cathode during electrodeposition, highly transparent tempered glass as the observation window and paraffin and hot melt adhesive for sealing after injection of electrolyte. Then the optically transparent cells were observed under a digital microscope camera (LEICA ICC50 W). The cross-sectional morphologies in depth were captured with the application of Ga$^+$ focused-ion beam at an accelerated voltage of 20 kV and 10 nA current (FIB, FEI Helios 650). All the Li electrodes unloaded from coin cells were washed with a mixed solvent of ethylene carbonate/dimethyl carbonate (EC:DMC = 1:1 vol%) and thoroughly dried before characterizations. All the characterizations were performed at the room temperature of 25 °C.

**Electrochemical measurements**. All the electrochemical tests were performed using CR2032 coin cells with a layer of commercial polypropylene (PP) separators (20 μm, Celgard 2250). The electrolyte used for electrochemical deposition experiments on MSC-Li and MPC-Li electrodes, optically transparent cells, symmetric cells and full cells was 1 M LiPF$_6$ in ethylene carbonate/dimethyl carbonate (EC:DMC = 1:1 vol%). In deposition experiments tests using perovskite-coated stainless steel foils as electrodes, electrolyte containing 1 M LiPF$_6$ in ethylene carbonate/dimethyl carbonate (EC:DMC = 1:1 vol%) with 2 vol% fluorinated ethylene carbonate (FEC) was applied. Supplementary Tables 1 and 2 is the data source of the box plots and error bars in Fig. 2b. For each row of the data, error bar stands for the upper and lower extremes of the range of values (UE, LE), the box stands for the 75 and 25 percentiles (UQ, LQ) and the square stands for the mean value of each set of data. In Li | LCO cells with 50 μm thick Li foil, electrolyte containing 1 M LiPF$_6$ in ethylene carbonate/dimethyl carbonate (EC:DMC = 1:1 vol%) with 1 vol% fluorinated ethylene carbonate (FEC) was applied. The Li$_4$Ti$_5$O$_{12}$ (LTO) cathodes used for full cell tests were made by blade-coating the N-methyl-2-pyrrolidone (NMP) slurry composed of 80 wt% Li$_4$Ti$_5$O$_{12}$ powder, 10 wt% super-P and 10 wt% poly(vinylidene difluoride) (PVDF) onto carbon-coated Al foil. The areal loading of active material is 3.2 mg cm$^{-2}$. The LiCoO$_2$ (LCO) cathodes were made by blade-coating the N-methyl-2-pyrrolidone (NMP) slurry composed of 80 wt% LiCoO$_2$ powder, 10 wt% super-P and 10 wt% poly(vinylidene difluoride) (PVDF) onto carbon-coated Al foil. The areal loading of active material is 20 mg cm$^{-2}$. Thin Li foil with thickness of 50 μm was purchased from Cellithium company. Electrochemical impedance spectroscopy (EIS) was tested on a VMP-3 (Biologic) electrochemical working station with a frequency range of 100 kHz to 100 mHz and an amplitude of 10 mV. Cyclic voltammetry tests were performed on a VMP-3 (Biologic) electrochemical working station with voltage ranges of 1 V to −0.1 V for perovskite-coated stainless steel foils at a scanning rate of 1 mV s$^{-1}$. Other electrochemical data including cycling tests of Li/Li symmetric cells and Li/LTO full cells was collected by a multi-channel cell testing station (CT2001A, Wuhan LAND). All the electrochemical measurements were conducted at a constant temperature of 25 °C.

## Data availability

The data that support the findings of this study are available from the authors on reasonable request, see author contributions for specific data sets.

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

## Acknowledgements

We acknowledge the funding support from the National Natural Science Foundation of China (Grants 51571184, 21875236 and 21805268), the Fundamental Research Funds for the Central Universities (Grant WK2060190085), and the joint Funds from Hefei National Synchrotron Radiation Laboratory (Grant KY2060000111). This work was partially carried out at the USTC Center for Micro and Nanoscale Research and Fabrication. G.Z.Z. and Q.W. are grateful to the Supercomputing Center of University of Science and Technology of China for the computing resource. We thank Haitao Liu for his assistance in FIB processing, and Xiu-Xia Wang and Jian Sun for their help with SEM observation.

## Author contributions

H.B.Y. and Y.C.Y. conceived and designed the experiments. G.Z.Z. and Q.W. performed simulations and calculations of Li transport mechanism in perovskite lattices. H.X.J. and Y.C.Y. performed XPS testing and analyzed the data. Y.C.Y., J.T.Y., C.H.J., F.L., H.S.M., and F.Z. carried out the fabrication of perovskite films, solid-state transfer, coin cell assembling and electrochemical tests. K.H.W. and J.S.Y. offered guidance of perovskite syntheses and characterizations. Y.C.Y. designed, assembled and tested the visible batteries and operated characterizations of SEM and EDX mappings. Y.C.Y., G.Z.Z. and Q.W. wrote the original draft. H.B.Y. and Y.C.Y. revised and further wrote the paper. All authors contributed to the data analysis. H.B.Y. directed the project.

## Competing interests

The authors declare no competing interests.
