## [Peer Review File · Nature Communications]

Reviewers' comments:

Reviewer #1 (Remarks to the Author):

The authors present a simple route to prepare two kinds of metal chloride perovskite thin films as solid electrolyte interfacial layers for lithium metal anodes. The perovskite thin films with low energy barriers facilitate fast Li⁺ ion transport, realizing dense deposition of lithium even at high current density of 5 mA/cm². The paper is well organized, but some points should be further clarified. A major revision is needed for further consideration.

1. In many cases, the perovskite shows poor stability. It is shown the Sn XPS peak was broadened after cycling. Thus, in this study, the stability of perovskite structure in the long cycling should be further discussed.
2. LTO usually has good cycling stability. However, in Fig.4a, a fast capacity decay was shown for the LTO battery after 500 cycles, which should be further explained.
3. The authors mentioned a Li-M alloy layer will be formed during a conversion-type electrochemical reaction, but the element M is not clear initially. More details should be added.
4. The authors mentioned "a gradient thin film consisting of perovskite framework on the top surface to isolate the liquid electrolyte and Li-M alloy layer". But according to the EDS mappings, the perovskite thin film is uniformly distributed along the cross section of the film.
5. The authors mentioned a Li-M alloy layer together with a LiCl layer are formed between the perovskite thin film and deposited lithium, more evidences are better to be provided.
6. If a Li-M alloy layer mixed with LiCl is formed, how can Li⁺ ions pass through this alloy layer, and this should be further explained.
7. The authors mentioned the perovskite thin films allow fast Li⁺ ion transport throughout its framework, the Li⁺ ion conductivities of the MASnCl₃ and MAPbCl₃ thin films need to be provided.

Reviewer #2 (Remarks to the Author):

The work by Hongbin Yao et.al fabricated a robust metal chloride perovskite thin film on the Li metal anode to isolate Li metal from liquid electrolyte in liquid lithium batteries. Although organic-inorganic halide perovskites have been used as anode materials in LIBs in recent reports, this work should be the first report about using such metal chloride perovskite as protective layer for Li anode. Taking advantages of spontaneous reaction between metal chloride perovskite and Li metal, a SEI layer can be in-situ formed on Li metal with fast Li⁺ transport. As reported in this article, an improved cycling stability can be realized by this interfacial strategy. To get this article qualified enough to be published on Nature Communications, however, there are still some points need to be addressed.

My detailed comments on the manuscript are given bellow:

1. The "solid electrolyte thin film" in the title is a little bit misleading since there's a big difference between "solid electrolyte thin film" formed on the surface of Li metal and real "solid electrolyte thin

film" fabricated by solid-state electrolyte.

2. Low energy barrier does not always indicate fast Li⁺ migration, what is the value of the real Li⁺ conductivity of the metal chloride perovskite? Furthermore, as indicated by the authors, Li⁺ intercalation and conversion reactions happened when metal chloride perovskite contacts with Li metal, how would the Li⁺ conductivity change?

3. Fig S.2a does not reflect the Li⁺ migration pathway through the lattice of perovskite, please modify that.

4. XPS depth profiling need to be further conducted to confirm the valence state of Sn or Pb not on the top surface and but also that directly contact with Li.

5. The (100), (200), and (300) diffraction peaks of lithiated perovskite shown in Fig 3c and Fig S9a are quite overlapped with that of the sealing bags, please re-consider the assignment of those three peaks by testing the XRD pattern of MSC-Li (before and after cycling) covered by films that do not show any sharp peaks (such as Kapton film).

6. The Ref. 35 presented higher-angle shift for XRD patterns of the lithiated perovskite electrode, which is opposite to the results showed here, please explain that. Another concern is that no other species due to the conversion and alloying reactions are reflected in the XRD patterns.

7. Some related references need to be cited, such as: *Electrochemical Energy Reviews* 1, no. 1 (2018): 35-53; *Electrochemical Energy Reviews* 2, no. 1 (2019): 1-28; *Angew. Chem. Int. Ed.* 2019, DOI: 10.1002/anie.201909805; *Advanced Materials* 30.45 (2018): 1804684.

Reviewers' comments:

Reviewer #1 (Remarks to the Author):

The authors present a simple route to prepare two kinds of metal chloride perovskite thin films as solid electrolyte interfacial layers for lithium metal anodes. The perovskite thin films with low energy barriers facilitate fast Li^+ ion transport, realizing dense deposition of lithium even at high current density of 5 mA/cm^2 . The paper is well organized, but some points should be further clarified. A major revision is needed for further consideration.

Reply: We are grateful to the referee for the nice comments, which are very helpful for improving the quality of our manuscript. We have addressed the following comments point by point and provided supplementary materials to support our conclusions. We hope that the revised manuscript is suitable for publication in Nature Communications.

Comment 1). In many cases, the perovskite shows poor stability. It is shown the Sn XPS peak was broadened after cycling. Thus, in this study, the stability of perovskite structure in the long cycling should be further discussed.

Reply 1: We thank the referee very much for bringing this to our attention.

The Sn XPS peak seems broadened after cycling mainly because of the weakened signal caused by the electrolyte residues on the top surface of perovskite-modified Li metal anode after cell cycling. These residues covering the perovskite layer would weaken the photoelectron intensity excited from the perovskite layer, which leads to a relatively low SNR (signal to noise ratio) and weak XPS signal of Sn. The weakened and broadened XPS peaks caused by the surface layer (SEI or oxide layer caused by the exposure to the air) were also observed in many previous works (e.g., Nat. Energy, 2017, 2(9): 17119., Adv. Mat., 2018, 30(36): 1705711., Joule, 2017, 1(4): 871-886.).

We confirmed the stability of perovskite structure after the cycling by XRD pattern analysis. We can see from the XRD patterns in Fig. 3c and Supplementary Fig. 25 in the revised manuscript that the diffraction peaks related to the (100) and (200) planes can be clearly observed after 200 h cycling. It means that the perovskite layer can maintain its crystalline phase and the framework structure after cycling.

Furthermore, we took the SEM and corresponding EDS mapping of MPC-Li in Li ||

LTO cell after 500 cycles at the rate of 5 C to show the good maintenance of perovskite layer on the surface of Li metal anode. As shown in Fig. R1a, most surface area of MPC-Li is smooth and dense after cycling, revealing the good stability of perovskite layer. The corresponding EDS mappings also show a uniform distribution of Pb and Cl (Fig. R1b and c), indicating a good stability of perovskite structure after the long cycling.

Fig. R 1 | Surface morphology of MPC-Li after being cycled for 500 cycles at 5 C and corresponding EDS mappings. a, SEM image of typical surface morphology of MPC-Li anode paired with LTO cathode after being cycled at 5 C for 500 cycles, **b-c**, corresponding EDX mappings indicating the distribution of Pb (**b**) and Cl (**c**). The scale bars are 50 μm.

Comment 2). LTO usually has good cycling stability. However, in Fig. 4a, a fast capacity decay was shown for the LTO battery after 500 cycles, which should be further explained.

Reply 2: The referee is acknowledged sincerely for this thoughtful comment.

Usually, the LTO based Li metal battery has good cycling stability when the ether electrolyte (e.g., 1 M LiTFSI in DOL:DME (v:v = 1:1)) is used (e.g., Nat. Energy, 2017, 2(9): 17119., Joule, 2017, 1(4): 871-886., Fig. R2a and b). As to the common carbonate electrolyte (e.g., 1 M LiPF₆ in EC:DMC (v:v = 1:1)), however, the cycling performance of Li metal || LTO cell is not as good as that using ether electrolyte. For

examples, as shown in Fig. R2c and d (corresponding to the data obtained from Adv. Energy Mater. 9, (2019), DOI: 10.1002/aenm.201901486 and ACS Nano 13.12, (2019): 14308-14318), when cycled at a low rate (1 C) and with a low LTO areal mass loading (2.0 mg cm^{-2}), the Li || LTO cell presents a capacity retention rate of approximately 80% after 400 cycles. Moreover, when the LTO areal mass loading increases to 3 mg cm^{-2} , the cycling performance of Li || LTO cell becomes worse with a capacity degradation of more than 50 % after 200 cycles (Fig. R2e, ACS Appl. Mater. Inter. 11, (2019): 11360-11368). For the high rate test, the Li || LTO cell exhibits a much poorer capacity retention with only 20 % capacity left after 300 cycles when a higher cycling rate of 2 C is applied (the black curve, Fig. R2f, ACS Appl. Mater. Inter. (2019), DOI: 10.1021/acsami.9b16083).

In our work, the testing conditions were even more rigorous with a high rate of 5 C, a high areal mass loading of 3.2 mg cm^{-2} and lean electrolyte (**Table R1**). Under a high current density of about 2.6 mA cm^{-2} , the surface of bare Li metal would become very loose, increasing the resistance and aggravate the side reactions between the electrolyte and Li metal. That's the reason why a fast capacity decay was shown for the bare Li metal || LTO battery after 500 cycles, as shown in Fig. 4a in the main text. In contrast, the cells using perovskite-modified Li metal anodes showed a greatly improved cycling stability, which indicates the effectiveness of our proposed metal halide perovskite film to protect Li metal and relieve the serious side reactions between the electrolyte and Li metal.

Fig. R2 | Cycling performance of Li || LTO cells reported in previous literatures. a-b, Cycling performance of Li || LTO cells using ether electrolyte (1 M LiTFSI in DOL:DME (v:v = 1:1)). **c-f,** Cycling performance of Li || LTO cells at different rate, with different cathode areal mass loadings and using different types of ester electrolyte and

different amount.

	Rate	Areal Mass Loading	Electrolyte Composition	Electrolyte Amount
c	1 C	2.0 mg cm ⁻²	1 M LiPF ₆ in EC:DEC (v:v = 1:1)	60 μl
d	1 C	2.0 mg cm ⁻²	1 M LiPF ₆ in EC:DEC (v:v = 1:1)	40 μl
e	1 C	3.0 mg cm ⁻²	1 M LiPF ₆ in EC:DMC:EMC (v:v:v = 1:1:1)	70 μl
f	2 C	2.6 mg cm ⁻²	1 M LiPF ₆ in EC:DEC (v:v = 1:1)	60 μl
this work	5 C	3.2 mg cm ⁻²	1 M LiPF ₆ in EC:DMC (v:v = 1:1)	50 μl

Table R1 | The comparison of different testing conditions between previous literatures and this work. The letters c, d, e and f refer to the examples shown in Fig. R2.

Comment 3). The authors mentioned a Li-M alloy layer will be formed during a conversion-type electrochemical reaction, but the element M is not clear initially. More details should be added.

Reply 3: We thank the referee very much for this valuable comment.

During the conversion-type electrochemical reaction, M²⁺ at the bottom of the perovskite film is reduced and combines with Li⁺ ions and electrons to form a Li-M alloy layer as shown in the following:

We further used the XPS to confirm the chemical state of the element M. Taking Li-Pb alloy layer for example, after an electrochemical deposition of 1 mAh cm⁻², which contains the conversion-type electrochemical reaction, the chemical state of the element Pb on the top surface of the perovskite layer was revealed to be Pb²⁺ (Supplementary Fig. 9c in the revised manuscript).

To expose the bottom part of the gradient layer, where the Li-Pb alloy composition lies, the top surface part was partly removed by an adhesive tape, as shown in Fig. R3a. In the Li-Pb alloy layer, the Pb 4f spectrum exhibits a main peak at 136.87 eV (Fig. R3b, yellow area), which can be assigned to Pb⁰ signal for the Li-Pb alloy. Another main peak at 138.01 eV (green area) can be attributed to the residual MAPbCl₃ and the grey peak at 134.50 eV can be assigned to P 2p coming from the electrolyte composition LiPF₆ during the initial electrochemical reaction process. Besides, the existence of alloyed Pb can be further evidenced by the signal of Li 1s at 55.18 eV, which indicates the presence of alloyed Li (Li-Pb alloy, Fig. R3c, green area). The side production LiCl exhibits its characteristic peak at 56.31 eV (blue area).

Little amount of exposed Li metal (53.41 eV) below implies the effectiveness of the adhesive tape stripping operation to expose the bottom part of the perovskite-alloy gradient layer. Furthermore, Cl 2p spectrum in Fig. R3d evidences the presence of MAPbCl₃ (200.38 eV) and LiCl (198.99 eV), which is in good agreement with the analyses of Fig. R3b and c, indicating the formation of alloy layer under the perovskite film.

We have added the above discussion in the revised manuscript (line 2-3, page 8 in main text, Supplementary Fig. 8, Supplementary Note 3 in Supplementary Information).

Fig. R3 | XPS analysis of the Li-M alloy layer formed during the conversion-type electrochemical reaction. a, schematic of the stripping operation via adhesive tape. **b-d,** XPS spectra of Pb 4f (**b**), Li 1s (**c**) and Cl 2p (**d**) after stripping the top part of the perovskite thin film.

Comment 4). The authors mentioned “a gradient thin film consisting of perovskite framework on the top surface to isolate the liquid electrolyte and Li-M alloy layer”. But according to the EDS mappings, the perovskite thin film is uniformly distributed along the cross section of the film.

Reply 4: We thank the referee for this thoughtful comment.

In our work, the concept “**gradient thin film**” refers to the gradient structure change of the protection layer from the perovskite on the top and the Li-M alloy at the bottom. Taking MSC-Li for an example, the chemical state of Sn is 2+ for MASnCl_3 on the top and 0 for Li-Sn alloy at the bottom. In other words, the unique structure of the gradient thin film can be proved via the observation of the chemical state difference of Sn in the top layer (Sn^{2+}) and the bottom layer (Sn^0).

It is hard to distinguish the gradient structure via EDS mappings because the EDS mapping is unable to characterize the chemical state of elements. That is why the perovskite thin film seems uniformly distributed along the cross section of the film according to EDS mappings without showing the gradient structure.

To show the gradient feature of the perovskite thin film on the Li metal anode, XPS analysis was conducted. We first used Ar ion sputtering to detect the element chemical state change via the depth profiling. But the perovskite materials might be easily damaged by the Ar ion beam. So we firstly checked the damage effect on pristine perovskite layers sputtered by Ar ions. For MASnCl_3 and MAPbCl_3 perovskite materials, chemical state of the B-site Pb^{2+} or Sn^{2+} ions in the perovskite was reduced to the metal (Pb^0 or Sn^0) induced by Ar ions. As shown in Fig. R4a, the prolongation of Ar ion sputtering time led to more reduction of Sn^{2+} to Sn^0 (red area). The situation is similar for Ar ion sputtered MAPbCl_3 as shown in Fig. R4b. Therefore, we think that the Ar ion sputtering is not suitable for depth analysis for our samples.

Fig. R4 | The reduction of Sn²⁺ and Pb²⁺ caused by Ar ion sputtering.

As an alternative way, we used adhesive tape to strip the perovskite thin film on the surface of Li metal anode for different times to check the chemical state change of the Sn or Pb (Fig. R5a). After the first stripping, the Sn 3d spectrum consists of 2 types of characteristic peaks (top, Fig. R5b). The yellow area exhibits a main peak at 484.19 eV (lower than the binding energy of ~ 484.8 eV of metallic Sn), which can be assigned to the Li-Sn alloy. Meanwhile, the blue signal area of the main peak is at 486.6 eV (Sn²⁺), which comes from the MASnCl₃. The results indicated the co-existence of Li-Sn alloy and MASnCl₃ in the exposed layer after the first time tape stripping. After sequent tape stripping for twice, the ratio of yellow area (Sn⁰) to blue area (Sn²⁺) of the main peaks increases obviously, indicating an increase of Li-Sn alloy composition and a decrease of MASnCl₃ part at a deeper layer. When the stripping time is up to 3, all XPS peaks can be assigned to the Li-Sn alloy and no peaks belonging to the MASnCl₃ can be observed. This means that all the Sn-contained compositions turn into the Li-Sn alloy at the bottom layer. Furthermore, MAPbCl₃ gradient layer exhibits the similar trend in composition change with the depth analysis implemented by the tape stripping. As shown in Fig. R5c, the chemical state change from Pb²⁺ to Pb⁰ indicates the composition transition from the MAPbCl₃ perovskite on the top surface to the Li-Pb alloy at the bottom layer. Together with the XPS results of the top surface of the perovskite layer as shown in the top part of Fig. R5b and c, the discussion and analyses above confirmed the perovskite-alloy gradient structure in the perovskite thin film on the Li metal anode.

We have clarified this point in the revised manuscript (line 13-16, page 10 in main text, Supplementary Fig. 16 and 17, Supplementary Note 7).

Fig. R5 | Depth XPS analysis of the perovskite thin films on the Li metal anodes to show the gradient composition change along the cross-section. a, schematic of the stripping operation via adhesive tape. **b-c**, XPS spectra comparison of Sn 3d (**b**), Pb 4f (**c**) at the top surface (the top pattern, data from Fig. 3d and Supplementary Fig. 28) and at different depth exposed by different times of the adhesive tape stripping (the bottom 3 patterns).

Comment 5). The authors mentioned a Li-M alloy layer together with a LiCl layer are formed between the perovskite thin film and deposited lithium, more evidences are better to be provided.

Reply 5: We thank the referee very much for this valuable comment.

The as-formed Li-M alloy layer with a LiCl layer is very thin and amorphous, so it is

hard to confirm their existence by XRD analysis. We used the XPS to confirm the existence of a Li-M alloy layer and a LiCl layer on the surface of the deposited Li metal. The presence of Li-M alloy layer and LiCl between perovskite and Li metal has been discussed and evidenced in the reply to the **Comment 3**. *All related evidences and discussions have been added into the revised manuscript (line 2-3, page 8 in main text, Supplementary Fig. 8, Supplementary Note 3 in Supplementary Information).*

Comment 6). If a Li-M alloy layer mixed with LiCl is formed, how can Li⁺ ions pass through this alloy layer, and this should be further explained.

Reply 6: We acknowledge this thoughtful comment by the referee.

Due to the electronic insulation property of the MAPbCl₃ and MASnCl₃ perovskite layer, the as-formed Li-M alloy layer mixed with LiCl at the interface is very thin and amorphous, which is beneficial for the Li⁺ ion to pass through. In addition, the chemical diffusion coefficients of lithium (D_{Li}) in the alloy layer is high (up to 10⁻⁴ cm² s⁻¹ for Li₂₂Sn₅ alloy layer), which can enable the conduction of Li⁺ ions in the formed interfacial layer. Meanwhile, the co-production of LiCl endows the composite layer with good electronic insulation, inducing the Li⁺ ions to pass through the composite layer. Provided with such condition, as reported by previous works including amorphous carbon and graphene as the protective film (Nat. Nanotech. 9, 618~623 (2014)., Chem. Mater. 27, 2780~2787 (2015)., Nano Lett. 14, 6016~6022 (2014).), the composite layer can build an electric field across the film, and provided a driving force for Li⁺ ions to migrate through the layer (Nat. Energy, 2017, 2(9): 17119.).

Comment 7). The authors mentioned the perovskite thin films allow fast Li⁺ ion transport throughout its framework, the Li⁺ ion conductivities of the MASnCl₃ and MAPbCl₃ thin films need to be provided.

Reply 7: We thank the referee for this valuable comment.

We adopted the perovskite/polymer composite films to reveal the Li⁺ ion conductivities in the MASnCl₃ and MAPbCl₃ films because it is hard to obtain free standing pure crystalline perovskite thin films for the ionic conductivity measurement. The composite films were fabricated via a simple solution-casting method. Poly(vinylidene fluoride) (PVDF) and corresponding chloride powder (LiCl, MAcl, and SnCl₂ (or PbCl₂)) were co-dissolved in the DMF (for MASnCl₃) or DMSO (for MAPbCl₃). The solutions were then casted and fully dried under high vacuum (< -0.1 Mpa) at 90 °C for 72 h to obtain the MASnCl₃-Li/PVDF and MAPbCl₃-Li/PVDF thin films with the polymer content of ~15% (wt%). To exclude the contribution of the halide ions (Cl⁻) migration to the ionic conductivity, the pristine MASnCl₃/PVDF

composite film and MAPbCl₃/PVDF composite film without adding LiCl were also fabricated. Moreover, the LiCl/PVDF composite film without perovskite was prepared to highlight the function of perovskite lattice in Li ion conductivity. All the composite films were pressed by stainless steel sheets at its 2 sides to be assembled in the 2032 cell for ionic conductivity measurement.

Fig. R6 | AC impedance spectroscopy of PVDF composite films with lithiated MASnCl₃, pristine MASnCl₃, lithiated MAPbCl₃, pristine MAPbCl₃ and LiCl. a, AC impedance of MASnCl₃-Li composite film. **b,** AC impedance of MASnCl₃

composite film. **c**, AC impedance of MAPbCl₃-Li composite film. **d**, AC impedance of MASnCl₃ composite film. **e**, AC impedance of LiCl composite film.

Composition	Area / cm ²	Thickness / mm	Impedance / Ω	Conductivity / (S cm ⁻¹)
MASnCl ₃ -Li	0.3	0.52	354.83	4.89 x 10 ⁻⁴
MASnCl ₃	0.3	0.47	167215.63	9.37 x 10 ⁻⁷
MAPbCl ₃ -Li	0.3	0.60	815.42	2.45 x 10 ⁻⁴
MAPbCl ₃	0.3	0.59	94875.17	2.07 x 10 ⁻⁶

$$\sigma = \text{Thickness} / (\text{Area} \times \text{Impedance})$$

Table R2 | Ionic conductivity of MASnCl₃-Li, MASnCl₃, MAPbCl₃-Li and MAPbCl₃ composite thin film.

The ionic conductivities of the fabricated composite films were tested via AC impedance (Fig. R6). As shown in Table R2, the Li⁺ ion conductivities of the MASnCl₃-Li/PVDF and MAPbCl₃-Li/PVDF films can be calculated as 4.89×10⁻⁴ and 2.45×10⁻⁴ S cm⁻¹, respectively (the 1st and 3rd rows, Table R2). In contrast, the ionic conductivities of the MASnCl₃/PVDF and MAPbCl₃/PVDF films are around 10⁻⁶ ~ 10⁻⁷ S cm⁻¹ (the 2nd and 4th rows, Table R2), much lower than those of MASnCl₃-Li/PVDF and MAPbCl₃-Li/PVDF films. Additionally, the cell with the LiCl/PVDF composite film turns out to be completely open-circuit with meaningless shape and pattern in its Nyquist plots (Fig. R6e), which further indicates the fast Li⁺ ion transport throughout the perovskite framework.

Reviewer #2 (Remarks to the Author):

The work by Hongbin Yao et.al fabricated a robust metal chloride perovskite thin film on the Li metal anode to isolate Li metal from liquid electrolyte in liquid lithium batteries. Although organic-inorganic halide perovskites have been used as anode materials in LIBs in recent reports, this work should be the first report about using such metal chloride perovskite as protective layer for Li anode. Taking advantages of spontaneous reaction between metal chloride perovskite and Li metal, a SEI layer can be in-situ formed on Li metal with fast Li^+ transport. As reported in this article, an improved cycling stability can be realized by this interfacial strategy. To get this article qualified enough to be published on Nature Communications, however, there are still some points need to be addressed.

Reply: We are grateful to the referee for the comprehensive and positive evaluation of our work.

My detailed comments on the manuscript are given bellow:

Comment 1). The "solid electrolyte thin film" in the tile is a little bit misleading since there's a big difference between "solid electrolyte thin film" formed on the surface of Li metal and real "solid electrolyte thin film" fabricated by solid-state electrolyte.

Reply 1: We thank the referee very much for the nice suggestion.

To avoid the misleading of the "solid electrolyte thin film", we have changed the title as "Metal chloride perovskite thin film based interfacial layer for shielding lithium metal from liquid electrolyte" in the revised manuscript.

Comment 2). Low energy barrier does not always indicate fast Li^+ migration, what is the value of the real Li^+ conductivity of the metal chloride perovskite? Furthermore, as indicated by the authors, Li^+ intercalation and conversion reactions happened when metal chloride perovskite contacts with Li metal, how would the Li^+ conductivity change?

Reply 2: We thank the referee very much for thoughtful comments.

We tested the Li^+ conductivity of the metal chloride perovskite with 25 % Li doping via AC impedance measurement. Perovskite/polymer composite films were adopted to reveal the Li^+ ion conductivities in the MASnCl_3 -Li films because it is hard to obtain free standing pure crystalline perovskite thin films for the ionic conductivity measurement. The composite films were fabricated via a simple solution-casting method. Poly(vinylidene fluoride) (PVDF) and corresponding chloride powder

(MACl, SnCl₂ and LiCl) were co-dissolved in the DMF. The solutions were then casted and fully dried under high vacuum (< -0.1 Mpa) at 90 °C for 72 h to obtain the MASnCl₃-Li/PVDF composite films with the polymer content of ~15% (wt%). To exclude the contribution of the halide ions (Cl⁻) migration to the ion conductivity, the pristine MASnCl₃/PVDF composite film without adding LiCl was also fabricated. Moreover, LiCl/PVDF composite film without perovskite was prepared to highlight the function of perovskite lattice in Li⁺ ion conductivity. All the composite films were pressed by stainless steel sheets at its 2 sides to be assembled in the 2032 cell for ionic conductivity measurement.

The ionic conductivities of the fabricated composite films were tested via AC impedance. As shown in Fig. R7a and Table R3 (the 1st row), the Li⁺ ion conductivity of the MASnCl₃-Li/PVDF film is calculated to be $4.89 \times 10^{-4} \text{ S cm}^{-1}$. In contrast, the ionic conductivity of the MASnCl₃/PVDF film is around $10^{-6} \sim 10^{-7} \text{ S cm}^{-1}$ (Fig. R7b and Table R3 (the 2nd row)), much lower than that of MASnCl₃-Li/PVDF film. Additionally, the cell with LiCl/PVDF composite film turns out to be completely open-circuit with meaningless shape and pattern in its Nyquist plots (Fig. R7c), which further indicates the fast Li⁺ ion transport throughout the perovskite framework.

Fig. R7 | AC impedance spectroscopy of PVDF composite films with lithiated MASnCl₃, pristine MASnCl₃ or LiCl. a, AC impedance of MASnCl₃-Li composite film. **b,** AC impedance of MASnCl₃ composite film. **c,** AC impedance of LiCl composite film.

Composition	Area / cm ²	Thickness / mm	Impedance / Ω	Conductivity / (S cm ⁻¹)
MASnCl ₃ -Li	0.3	0.52	354.83	4.89 x 10 ⁻⁴
MASnCl ₃	0.3	0.47	167215.63	9.37 x 10 ⁻⁷

$$\sigma = \text{Thickness} / (\text{Area} \times \text{Impedance})$$

Table R3 | Ionic conductivity of MASnCl₃-Li and MASnCl₃ perovskite composite film.

When applying the metal chloride perovskite thin film onto the Li metal anode, once the metal chloride perovskite thin film and Li metal were pressed together, the reduction and alloying reaction occurred at the interface due to the high reduction potential of Li metal. The as-formed Li-M alloy layer mixed with LiCl at the interface is very thin and amorphous, which is beneficial for the Li⁺ ion to pass through. In addition, the intercalation process of Li⁺ ion into the perovskite framework is calculated (Fig. R8a and b). This intercalation process is thermodynamically downhill, as evidenced by the release of heat of 4.88 eV as one Li⁺ ion intercalates into the MASnCl₃ perovskite framework (Fig. R8c), indicating a strong spontaneous trend for Li⁺ ion to intercalate into the metal chloride perovskite. Therefore, as an interfacial layer on the surface of Li metal anode, the good Li⁺ ionic conductivity can be retained in the metal chloride perovskite thin films, which is also confirmed by the AC impedance tests in the symmetric cells (Fig. 3a).

We have included above discussion in the revised manuscript (line 13-14, page 10 in the main text and Supplementary Fig. 15, Supplementary Note 6 in Supplementary Information).

Fig. R8 | Structure and energy calculation of the pristine and lithiated metal chloride perovskite. **a**, Structure of MASnCl₃ lattice (top) and MASnCl₃-Li lattice (bottom). **b**, the table of the lattice parameters of MASnCl₃ and MASnCl₃-Li. **c**, the table of energy calculation of the Li⁺ ion intercalation behavior.

Comment 3). Fig S.2a does not reflect the Li⁺ migration pathway through the lattice of perovskite, please modify that.

Reply 3: We thank the referee for bringing this to our attention.

We have modified the Fig S.2a in the revised version of Supplementary Information to ensure the preciseness of the schematic, as shown in Fig. R9.

Fig. R9 | Li⁺ migration pathway through the lattice of perovskite.

Comment 4). XPS depth profiling need to be further conducted to confirm the valence state of Sn or Pb not on the top surface and but also that directly contact with Li.

Reply 4: We thank the referee very much for this valuable comment.

To perform the XPS depth profiling, we first tried Ar ion sputtering to processing the depth analysis. But the perovskite materials were easily damaged by the Ar ion beam. For the MASnCl_3 or MAPbCl_3 perovskite material, chemical state of the B-site Pb^{2+} or Sn^{2+} ions in the perovskite was reduced to the zero (Pb^0 or Sn^0) induced by Ar ions. As shown in Fig. R10a, the prolongation of Ar ion sputtering time leads to more reduction of Sn^{2+} to Sn^0 (red area). The situation is similar for Ar ion sputtered MAPbCl_3 as shown in Fig. R10b. Therefore, we think that the Ar ion sputtering is not suitable for depth analysis for our samples.

Fig. R10 | The reduction of Sn^{2+} and Pb^{2+} in the perovskite thin film caused by Ar ion sputtering.

As an alternative way, we used adhesive tape to strip the perovskite thin film on the surface of Li metal anode for different times to check the chemical state change of the Sn or Pb (Fig. R11a). After the first time tape stripping, the Sn 3d spectrum consists of 2 types of characteristic peaks (top, Fig. R11b). The yellow area exhibits a main peak at 484.19 eV (lower than the binding energy of ~ 484.8 eV of metallic Sn), which can be assigned to the Li-Sn alloy. Meanwhile, the blue signal area of the main peak is at 486.6 eV (Sn^{2+}), which comes from the MASnCl_3 . The results indicate the co-existence of Li-Sn alloy and MASnCl_3 in the exposed layer after the first stripping. After sequent stripping for twice, the ratio of yellow area (Sn^0) to blue area (Sn^{2+}) of

the main peaks increases obviously, indicating an increase of Li-Sn alloy composition and a decrease of MASnCl_3 part at a deeper layer. When the stripping time is up to 3, all XPS peaks can be assigned to the Li-Sn alloy and no peaks belonging to the MASnCl_3 can be observed. This means that all the Sn-contained compositions turn into the Li-Sn alloy at the bottom layer. Furthermore, MAPbCl_3 gradient layer exhibits the similar trend in composition change with the depth analysis implemented by the tape stripping. As shown in Fig. R11c, the chemical state change from Pb^{2+} to Pb^0 indicates the composition transition from the MAPbCl_3 perovskite on the top surface to the Li-Pb alloy at the bottom layer. Together with the XPS results of the top surface of the perovskite layer as shown in the top part of Fig. R11b and c, the discussion and analyses above confirmed the perovskite-alloy gradient structure in the perovskite thin film on the Li metal anode.

Fig. R11 | Depth XPS analysis of the perovskite thin films on the Li metal anodes to show the gradient composition change along the cross-section. a, schematic of the stripping operation via adhesive tape. **b-c,** XPS spectra comparison of Sn 3d (**b**), Pb 4f (**c**) at the top surface (the top pattern, data from Fig. 3d and Supplementary Fig. 28) and at different depth exposed by different times of the adhesive tape stripping (the bottom 3 patterns).

In addition, the tape stripping for 3 times is sufficient to expose the layer which directly contact with the Li metal. Taking MPC-Li for an example, after stripping for 3 times, the Li 1s spectrum exhibits 3 types of characteristic peaks (Fig. R12). The red area and the green area can be assigned to the presence of LiCl and Li-Pb alloy, respectively. The small blue area with the characteristic peak at 53.82 eV belongs to Li metal, indicating that after stripping for 3 times, the element Pb directly contacting with Li metal is exposed.

We thank the referee again for this valuable comment and have added the discussion and analysis above into the revised manuscript (line 14-16, page 10 in the main text and Supplementary Fig. 16, 17 and 18, Supplementary Note 7 in Supplementary Information).

Fig. R12 | Li 1s XPS spectrum of MPC-Li after stripping for 3 times.

Comment 5). The (100), (200), and (300) diffraction peaks of lithiated perovskite shown in Fig 3c and Fig S9a are quite overlapped with that of the sealing bags, please re-consider the assignment of those three peaks by testing the XRD pattern of MSC-Li (before and after cycling) covered by films that do not show any sharp peaks (such as Kapton film).

Reply 5: We thank the referee very much for this thoughtful comment.

We re-checked and re-considered the original XRD results. We made a control XRD test that putting the pristine perovskite-coated substrate in the sealing bag and astonishingly found that the diffraction peaks at around 9° , 18° and 27° are even strengthened (Fig. R13, red line). Meanwhile, the sealing bag was found to seriously weaken the intensity of pristine perovskite XRD peaks, making them hard to be observed (Fig. R13, blue line and insets). Thus, we think that the strong (100), (200),

and (300) diffraction peaks of lithiated perovskite shown in the original Fig 3c and Fig S9a in the manuscript are resulted by the sealing bag.

Fig. R13 | XRD patterns and some confusing peaks caused by the plastic sealing bag.

To clarify the XRD peaks of the lithiated perovskite, we used a customized XRD testing device with the as-mentioned Kapton film for sealing (Fig. R14a), which does not show any sharp peaks. The results show that the lithiated perovskite's diffraction peaks of (100) and (200) are located at around 15.5° and 31° , which can be clearly observed before and after 100 cycles (Fig. R14b and c). *We have re-tested the XRD for all corresponding samples and replaced the original XRD data with new ones in the revised manuscript (Fig. 3c) and supplementary material (Supplementary Fig. 10, 25 and 26).*

Fig. R14 | Updated XRD patterns of lithiated perovskite using new sealing device without any misleading peaks. a, Schematic of the sealing device for XRD testing, **b,** XRD patterns of lithiated MASnCl₃ before and after 100 cycles, **c,** XRD patterns of lithiated MAPbCl₃ before and after 100 cycles.

Comment 6). The Ref. 35 presented higher-angle shift for XRD patterns of the lithiated perovskite electrode, which is opposite to the results showed here, please explain that. Another concern is that no other species due to the conversion and alloying reactions are reflected in the XRD patterns.

Reply 6: We thank the referee very much for the valuable comments.

According to the results of the retested XRD patterns of the lithiated perovskite thin film (red line, Fig. R15a and b), the series of (100) and (200) plane diffraction peaks

did not show obvious shift in comparison to that of the raw perovskite thin film (blue line, Fig. R15 a and b). This phenomenon is in good agreement with previous reports that the intercalation of Li ion does not cause obvious change of the perovskite framework (Fig. R16, *J. Mater. Chem. A*, 2019, 7(21): 13043-13049., *Adv. Sci.*, 2018, 5(12): 1800736.). In this context, it is hard to be understood that the XRD peaks shifted to higher angle as mentioned in the Ref. 35 because this shift indicates the shrinkage of the perovskite lattice, which can not happen with the ion intercalation into the perovskite framework. Therefore, we removed the ref. 35 in the revised manuscript.

Fig. R15 | Comparison of XRD patterns of lithiated perovskite and raw perovskite. **a**, XRD patterns of lithiated MASnCl_3 (red) and raw MASnCl_3 (blue), **b**, XRD patterns of lithiated MAPbCl_3 (red) and raw MAPbCl_3 (blue). The black line belongs to the XRD pattern of the sealing device.

Fig. R16 | XRD patterns of Li-doped perovskite. **a**, FAPbCl_3 nanostructures grown with LiCl additive (red, doping ratio: 10%), without LiCl additive (blue). **b**, XRD patterns of a series of Li-doped MAPbI_3 films with different Li-doping ratio.

The species of the conversion and alloying reactions are mainly Li-M alloy and LiCl, which are amorphous and hard to be detected by the XRD analysis. We used the XPS analysis to reveal the compositions of the formed species. In the reply to **the comment 4**, we have confirmed the existence of Li-M alloy and LiCl at the interface between the perovskite thin film and Li metal anode.

We thank the referee again for this valuable comment and have added the discussion and analysis above into the revised manuscript (line 22-27, page 12 in main text, Supplementary Fig. 26 and 27 in Supplementary Information).

Comment 7). Some related references need to be cited, such as: Electrochemical Energy Reviews 1, no. 1 (2018): 35-53; Electrochemical Energy Reviews 2, no. 1 (2019): 1-28; Angew. Chem. Int. Ed. 2019, DOI: 10.1002/anie.201909805; Advanced Materials 30.45 (2018): 1804684.

Reply 7: We thank the referee for this valuable comment.

We have added all the literatures mentioned above to the reference list in the revised manuscript (*Reference number: 4, 6, 20 and 22*).

REVIEWERS' COMMENTS:

Reviewer #1 (Remarks to the Author):

In this version, the authors well replied to the comments and made the corresponding modifications.

Reviewer #2 (Remarks to the Author):

I carefully checked the reply from the authors. I think that the authors have addressed very well comments and suggestions. It is publishable now.

Reviewers' comments:

Reviewer #1 (Remarks to the Author):

In this version, the authors well replied to the comments and made the corresponding modifications.

Reply 1: We the referee for his/her positive response and nice comments of first round review for improving the quality of our manuscript.

Reviewer #2 (Remarks to the Author):

I carefully checked the reply from the authors. I think that the authors have addressed very well comments and suggestions. It is publishable now.

Reply 2: We thank the referee for their recognition and positive evaluation of our revision. We highly appreciate the time and expertise provided by the referee.

Acknowledgement:

Here we make acknowledgements according to the requests by the publishers from which we got the permission/license to reuse and reproduce the images.

Fig. R2 a: Reprinted from Joule, 1(4), Pang Q, Liang X, Shyamsunder A, et al., An in vivo formed solid electrolyte surface layer enables stable plating of Li metal[J], 871-886., 2017, with permission from Elsevier.

Fig. R2 b: Reprinted by permission from [Springer Nature]: [Nature] [Nature Energy] [A facile surface chemistry route to a stabilized lithium metal anode, Xiao Liang et al [2017].

Fig. R2 c: Reprinted from Pathak R, Chen K, Gurung A, et al. Ultrathin bilayer of graphite/SiO₂ as solid interface for reviving Li metal anode[J]. Advanced Energy Materials, 2019, 9(36): 1901486., with permission from John Wiley and Sons.

Fig. R2 d: Reprinted (adapted) with permission from (Shi H, Zhang C J, Lu P, et al. Conducting and Lithiophilic MXene/Graphene Framework for High-Capacity, Dendrite-Free Lithium–Metal Anodes[J]. ACS nano, 2019, 13(12): 14308-14318.). Copyright (2019) American Chemical Society."

Fig. R2 e: Reprinted (adapted) with permission from (Ouyang Y, Guo Y, Li D, et al. Single Additive with Dual Functional-Ions for Stabilizing Lithium Anodes[J]. ACS applied materials & interfaces, 2019, 11(12): 11360-11368.). Copyright (2019) American Chemical Society."

Fig. R2 f: Reprinted (adapted) with permission from (Li K, Wang Y, Jia W, et al. Polymer Electrolyte Film as Robust and Deformable Artificial Protective Layer for High-Performance Lithium Metal Anode[J]. ACS Applied Materials & Interfaces, 2019.). Copyright (2019) American Chemical Society."

Fig. R16 a (Supplementary Fig. 27 a): Reproduced from Gong J, Li X, Guo P, et al. Energy-distinguishable bipolar UV photoelectron injection from LiCl-promoted FAPbCl₃ perovskite nanorods[J]. Journal of Materials Chemistry A, 2019, 7(21): 13043-13049., with permission from the Royal Society of Chemistry.

Fig. R16 b (Supplementary Fig. 27 b): Reproduced from Fang Z, He H, Gan L, et al. Understanding the Role of Lithium Doping in Reducing Nonradiative Loss in Lead Halide Perovskites[J]. Advanced Science, 2018, 5(12): 1800736.. This is an open access article under the terms of the Creative Commons Attribution License, which permits use, distribution and reproduction in any medium, provided the original work is properly cited.